# Nox4 expression in osteo-progenitors controls bone development in mice during early life

Jin-Ran Chen [1,2 ✉], Oxana P. Lazarenko[1,2], Michael L. Blackburn[1,2], Jennifer F. Chen[3], Christopher E. Randolph[4], Jovanny Zabaleta[5], Katrin Schroder [6], Kim B. Pedersen[7] & Martin J. J. Ronis [7 ✉]

Tightly regulated and cell-specific NADPH-oxidases (Nox) represent one of the major sources of reactive oxygen species (ROS) signaling molecules that are involved in tissue development and stem cell self-renewal. We have characterized the role of Nox4 in osteo-progenitors during postnatal bone development. Nox4 expression in bone and ROS generation were increased during early osteoblast differentiation and bone development. Stromal osteoblastic cell self-renewal, proliferation and ROS production were significantly lower in samples from whole-body Nox4 knockout mice (Nox4$^{-/-}$) and conditional knockout (CKO) mice with depletion of Nox4 in the limb bud mesenchyme compared with those from control mice (Nox4$^{fl/fl}$), but they were reversed after 9 passages. In both sexes, bone volume, trabecular number and bone mineral density were significantly lower in 3-week old CKO and Nox4$^{-/-}$ mice compared with Nox4$^{fl/fl}$ controls. This was reflected in serum levels of bone formation markers alkaline phosphatase (ALP) and procollagen 1 intact N-terminal propeptide (P1NP). However, under-developed bone formation in 3-week old CKO and Nox4$^{-/-}$ mice quickly caught up to levels of control mice by 6-week of age, remained no different at 13-week of age, and was reversed in 32-week old male mice. Osteoclastogenesis showed no differences among groups, however, CTX1 reflecting osteoclast activity was significantly higher in 3-week old male CKO and Nox4$^{-/-}$ mice compared with control mice, and significantly lower in 32-week old Nox4$^{-/-}$ mice compared with control mice. These data suggest that Nox4 expression and ROS signaling in bone and osteoblastic cells coordinately play an important role in osteoblast differentiation, proliferation and maturation.

[1] Arkansas Children's Nutrition Center, Little Rock, AR 72202, USA. [2] Department of Pediatrics, University of Arkansas for Medical Sciences, Little Rock, AR 72202, USA. [3] Undergraduate Pre-Medical Program, University of Arkansas at Fayetteville, Fayetteville, AR 72701, USA. [4] Center for Translational Pediatric Research, Arkansas Children's Research Institute, Little Rock, AR 72202, USA. [5] Department of Pharmacology and Experimental Therapeutics, Louisiana State University Health Sciences Center, New Orleans, LA 70112, USA. [6] Institute of Physiology I, Goethe-University, Frankfurt, Germany. [7] Department of Interdisciplinary Oncology (DIO), Stanley S. Scott Cancer Center, Louisiana State University Health Sciences Center, Louisiana Cancer Research Center, New Orleans, LA 70112, USA. ✉email: chenjinran@uams.edu; mronis@lsuhsc.edu

Bone is a dynamic organ with bone formation and resorption changing during the course of development as the skeleton undergoes age-dependent regulation. Osteoblastic bone formation is a dominant event during early life bone modeling, leading to the majority of bone mass to be deposited before puberty[1]. However, in adults, fine bone remodeling is a process in which osteoblasts and osteoclasts are coupled to maintain bone mass at an appropriate level[2]. It has been suggested that bone mass steadily declines as early as in young adulthood[3], and thereafter, significant bone loss occurs during aging largely because of decreasing osteoblast number and bone formation rate[4]. Significant bone loss during aging is accompanied by increases in the generation or accumulation of reactive oxygen species (ROS)[5]. On the other hand, appropriate concentrations of ROS may be required for optimal early tissue development since it has been shown that ROS serve as signaling molecules controlling cell differentiation and proliferation[6,7].

We have previously shown that osteoblast differentiation and proliferation are unique processes, which are dependent on the status of the cell senescence signaling pathway, i.e., increased signaling through the senescence pathway prevents a cell from proliferating and restricts the capacity of a stromal cell to differentiate into an active osteoblast[8]. However, maintaining a balance between ROS production, elimination of excessive ROS and senescence seems to be essential for physiological bone homeostasis at any age. During the past several decades, much have been established regarding the potential harmful effects of increased levels of ROS to damage DNA, lipids and proteins leading to many pathophysiologic conditions, including degenerative bone diseases. However, much less work has examined the skeletal consequences of reduced ROS production[9]. Previous study suggested that a reduction in ROS levels and signaling may be detrimental to bone development[10]. Moreover, a previous study indicated that decreased ROS concentrations actually promoted inflammation in the aging musculoskeletal system[11].

ROS can be generated by several sources. The tightly controlled and cell-specific NADPH-oxidases (Nox) represent one of the major sources of intracellular ROS signaling molecules, including superoxide, hydrogen peroxide, and hydroxyl radicals in many cell types. Activation mechanisms, tissue distribution and subcellular localization of different members of the Nox family are markedly different[12]. Given that they are macrophages by nature, bone-resorptive osteoclasts are expected to express all subtypes of Nox[13]. Therefore, it was not surprising that the majority of ROS generated in osteoclasts is through a Nox-dependent mechanism[14,15]. RANKL and TNFα have been shown to stimulate ROS generation in osteoclasts or their precursors[16,17]. However, it was unexpected that bone resorption in Nox2-deficient mice was reported to be apparently normal during early development[11]. We have previously identified that osteoblastic cells do not express Nox3 but do express Nox 1, 2 and 4, with isoforms 4 and 2 being relatively abundant[18]. We have characterized the role of Nox2 during bone development by using a mouse model in which its co-factor p47phox was knocked out[19]. We have also reported recently that systemic deletion of Nox4 had minimum effects on age- or OVX-induced bone loss in adult female mice[20]. However, additional investigation of the tissue protective activity of NADPH-oxidases (Nox) during development, the origins and dynamics of ROS production, and their critical biological targets in bone are required.

We have previously hypothesized that Nox subtype-associated ROS production could have signaling pathways acting independently or synergistically to control tissue and cell development. Subtypes of Nox may generate physiological levels of ROS for tissue development to compensate for loss of one subtype of Nox. For example, Nox2-deficient (or deficiency of its components such as p47phox) mice are animal models for chronic granulomatous disease (CGD). Normal bone resorption in CGD patients and in Nox2-deficient mice might be due to partial compensation of ROS production by up-regulation of other isoforms of Nox such as Nox4 in bone cells[9,21]. However, animals with deletion of Nox2 still have age-related joint destruction[22]; indicating that Nox2 signaling does have a role in controlling local tissue pathophysiology. It is not known if systemic or if tissue specific deletion of Nox4 might have consequences on early postnatal bone development. In the present study, we set out to answer some of these critical questions in bone and focus on osteoblast differentiation and postnatal bone development in early life before weaning and in adults using mouse models with global Nox4 knockout or with pre-osteoblast-specific Nox4 gene deletion in the long bones.

## Results

### Changes of Nox expression and ROS production during bone cell differentiation in vitro

We examined cell proliferation, ROS production and Nox expression during in vitro osteoblast and osteoclast differentiation. In cell cultures of the murine osteoblast stromal cell line ST2, it took 7 days for cells to be fully differentiated into alkaline phosphatase (ALP)-positive mature osteoblasts in the presence of osteoblast culture medium (Fig. 1a). During this period, cell proliferation gradually increased from day 1 to day 7 (Fig. 1b) with ROS production reaching a peak at day 3 (Fig. 1c). In cell cultures of the murine macrophage cell line RAW264.7, osteoclastogenesis was induced by 30 ng/ml RANKL. It took 5 days to differentiate the progenitors to TRAPase-positive mature multi-nuclear osteoclasts (Fig. 1d). During this osteoclast differentiation period, cell proliferation significantly increased from day 1 to day 3 (Fig. 1e). ROS production dramatically increased from day 1 to day 3, and significantly decreased from day 3 to day 5 (Fig. 1f). Elevated ROS production thus seems to be a feature of both early osteoblastic and early osteoclastogenic differentiation, at least in vitro.

During osteoblastic differentiation, Nox4 mRNA expression was significantly higher at day 3 than at the other time points (Fig. 1g). On the other hand, Nox1 and Nox2 mRNA expressions were highest at day 1 (Fig. 1h, i). Like ALP-staining, the mRNA expression of the osteoblast differentiation marker ALP was dramatically increased on day 7 compared to earlier time points (Fig. 1j). Thus, ALP expression increases while Nox4 expression decreases.

### Bone marrow stromal cells from Nox4 knockout mice have reduced ability to differentiate into osteoblasts

We isolated bone marrow stromal cells from adult female mice for in ex vivo cultures. The mice were of three Nox4 genotypes: Global Nox4 knockout mice (Nox4⁻/⁻), Nox4 CKO mice of genotype PrxCre⁺/⁻ Nox4ᶠˡ/ᶠˡ with depletion of Nox4 in cells of the chondro/osteoblastic lineage in the long bones and control mice (Nox4ᶠˡ/ᶠˡ). There were three mice per genotype. We found those stromal cells from Nox4 CKO and Nox4⁻/⁻ mice appeared to have decreased proliferation than cells from Nox4ᶠˡ/ᶠˡ control mice (Fig. 2a) with reduced ability to form stem cell units (Fig. 2b). After 12 days of cultures in the presence of osteoblast differentiation culture medium, numbers of colony-forming units of fibroblasts (CFU-Fs, more than thirty cells considered as one colony) were significantly lower in cell cultures from CKO and Nox4⁻/⁻ mice compared with cell cultures from Nox4ᶠˡ/ᶠˡ control mice ($p < 0.05$) (Fig. 2c). Significantly reduced stromal cell proliferation was in cells from CKO and Nox4⁻/⁻ mice compared with cells from Nox4ᶠˡ/ᶠˡ control mice persisted in passage 2 of the cells (Fig. 2d). Interestingly, senescence-associated β-galactosidase (SAβG) activity was significantly lower in cells from CKO and

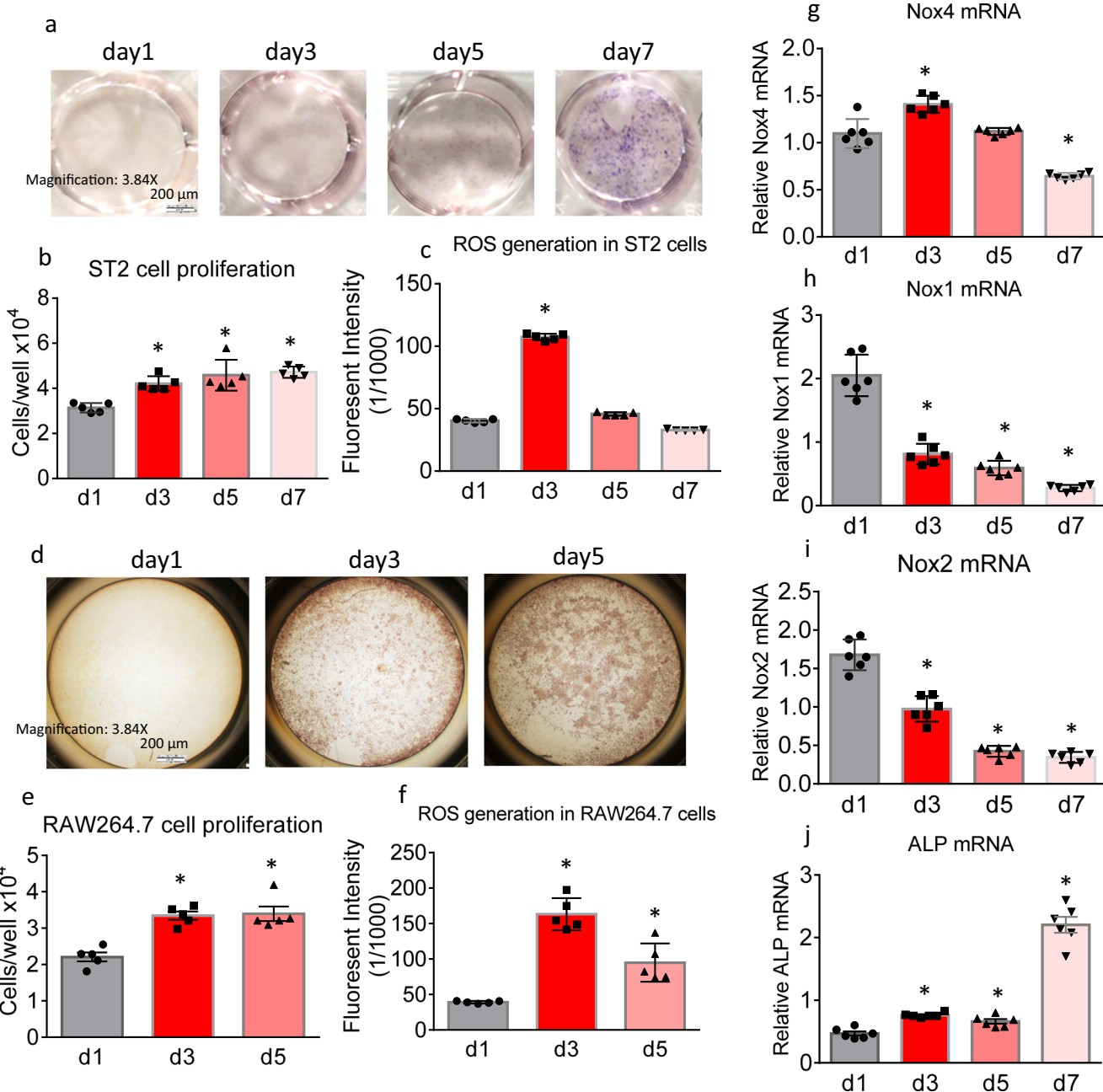

**Fig. 1 Nox expression and ROS production in bone cells during their differentiation. a** Representative cell culture pictures of ALP staining of ST2 cells in the osteoblast differentiation medium on day 1, day 3, day 5, and day 7. **b** ST cell proliferation in the osteoblast differentiation medium on day 1, day 3, day 5, and day 7. **c** ROS generation from ST2 cells cultured in the osteoblast differentiation medium on day 1, day 3, day 5, and day 7. **d** Representative cell culture pictures of TRAPase staining of RAW264.7 cells in the presence of 30 ng/ml of RANKL on day 1, day 3, and day 5. **e** RAW264.7 cell proliferation on day 1, day 3, and day 5 in the presence of 30 ng/ml of RANKL. **f** ROS generation from RAW264.7 cells cultured in the presence of 30 ng/ml of RANKL on day 1, day 3, and day 5. Real-time PCR determined (**g**) Nox4, (**h**) Nox1, (**i**) Nox2 and (**j**) ALP mRNA expression in ST2 cells cultured in osteoblast differentiation medium on day 1, day 3, day 5, and day 7, six culture wells of 12-well culture plates per group. *$p < 0.05$ by one-way ANOVA followed by Tukey's post hoc test compared with Day 1. For each cell line, the results are from one cell-culture experiment with each treatment given to 6 wells.

$Nox4^{-/-}$ mice compared with cells from $Nox4^{fl/fl}$ control mice (Fig. 2e).

We have also examined the osteoclastogenic activity of non-adherent bone marrow hematopoietic cells. Osteoclast differentiation were apparently not different from hematopoietic cells isolated from the three Nox4 genotypes (Fig. 3a, b). In the ex vivo cultures in the presence of RANKL, multi-nuclear TRAPase-positive osteoclast number per well showed no differences among groups (Fig. 3c). Expression of two osteoclast differentiation

associated genes, NFATc1 and NFκB, in cells derived from non-adherent bone marrow hematopoietic cells were not significantly different among groups (Fig. 3d, e). It is known that β-Catenin expression is sensitive to the changes of ROS production. Its mRNA expression was significantly lower in osteoclastic cultures from CKO and $Nox4^{-/-}$ mice compared with cells from $Nox4^{fl/fl}$ mice (Fig. 3f).

We continued growing the bone marrow stromal cell cultures from Nox4 CKO, $Nox4^{-/-}$ and $Nox4^{fl/fl}$ control mice, and passaged

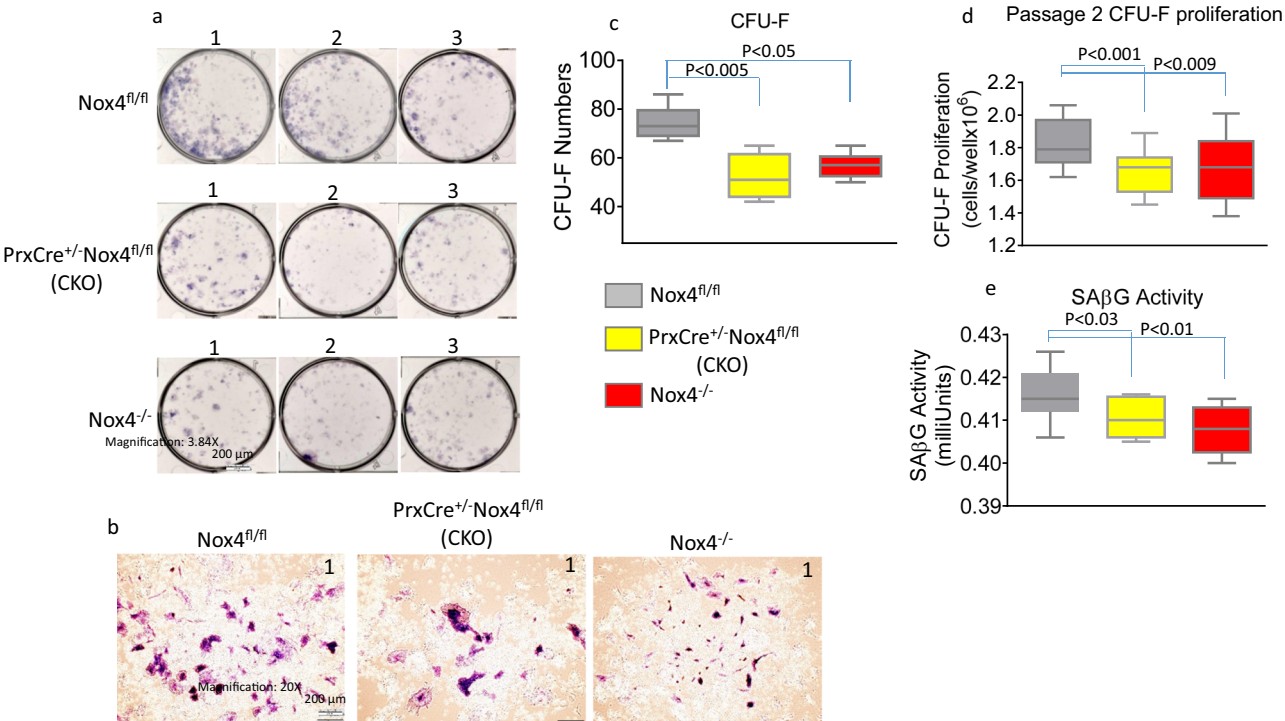

**Fig. 2 Bone marrow stromal cells from CKO and Nox4⁻/⁻ mice have reduced ability to proliferate and self-renew.** Bone marrow stromal cells were isolated from femurs of 18–31-weeks old female mice of genotypes *Nox4ᶠˡ/ᶠˡ* control, *PrxCre⁺/⁻ Nox4ᶠˡ/ᶠˡ* CKO, and *Nox4⁻/⁻* and cultured. **a** Representative pictures of hematoxylin blue stained cell culture wells from each mouse. Numbers on top of each well are experimental animal numbers. **b** A typical single stem cell colony after 12 days cultures of bone marrow stem cells from mice of the three genotypes. **c** Numbers of colony-forming units of fibroblasts (CFU-F) after 12 days cultures of bone marrow stem cells from mice of the three genotypes. **d** Passage 2 colony-forming units of fibroblast proliferation of cells from mice of the three genotypes. **e** Senescence-associated β-galactosidase (SAβG) activity of passage 2 colony-forming units of fibroblasts. Data bars are expressed as mean ± SD (*n* = 3/group). Significant differences indicated by *p* < 0.05, by one-way ANOVA followed by Tukey's post hoc test.

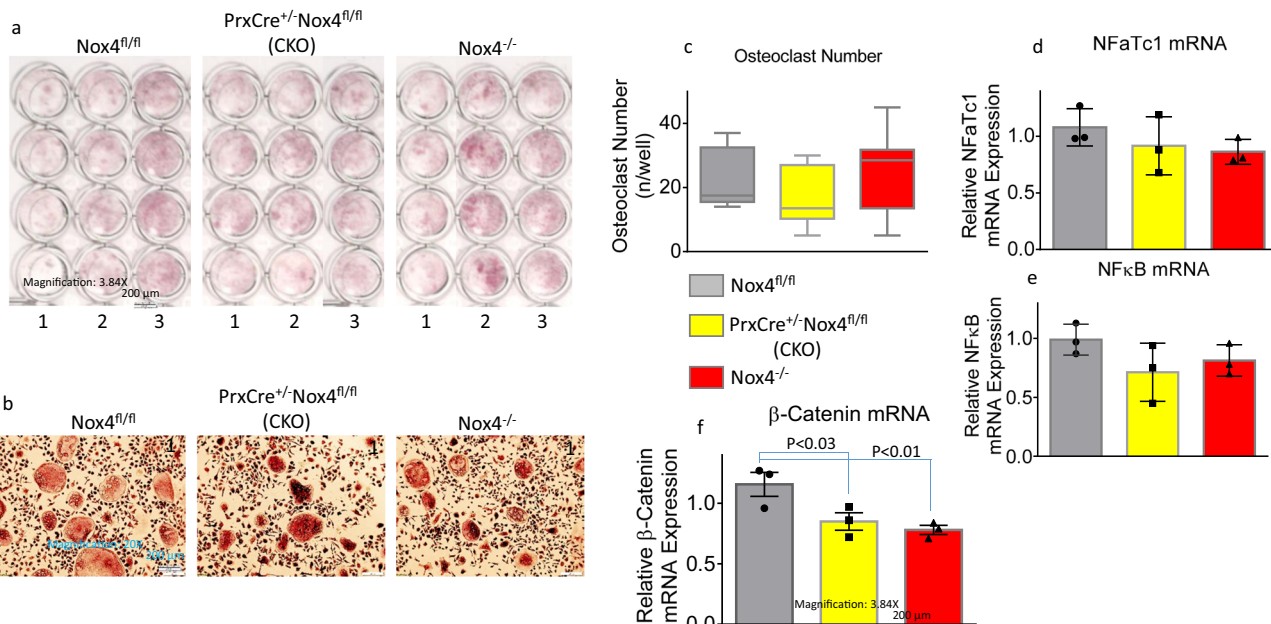

**Fig. 3 Nox4 gene deletion has minimal effects on osteoclastogenesis. a** Non-adherent bone marrow cells were cultured in the presence of 30 ng/ml RANKL for 6 days. Pictures are showing TRAPase staining of cells from *Nox4ᶠˡ/ᶠˡ* control, *PrxCre⁺/⁻ Nox4ᶠˡ/ᶠˡ* CKO and *Nox4⁻/⁻* mice with four wells of 96-well plat for cells isolated from each mouse, numbers on bottom of each well are experimental animal numbers. **b** A typical area of TRAPase pink stained multi-nuclear osteoclasts from non-adherent bone marrow cells from each of the three genotypes. **c** TRAPase-positive multi-nuclear osteoclasts were counted from each well (four wells per mouse cells) of cultures from each Nox4 genotype. **d–f** Real-time PCR shows relative mRNA expression of NFaTc1, NFκB, and β-Catenin in total RNA isolated from cultures of non-adherent bone marrow cells from *Nox4ᶠˡ/ᶠˡ* control, *PrxCre⁺Nox4ᶠˡ/ᶠˡ* CKO, and *Nox4⁻/⁻* mice, three mice per group. Significant differences indicated by *p* < 0.05, by one-way ANOVA followed by Tukey's post hoc test.

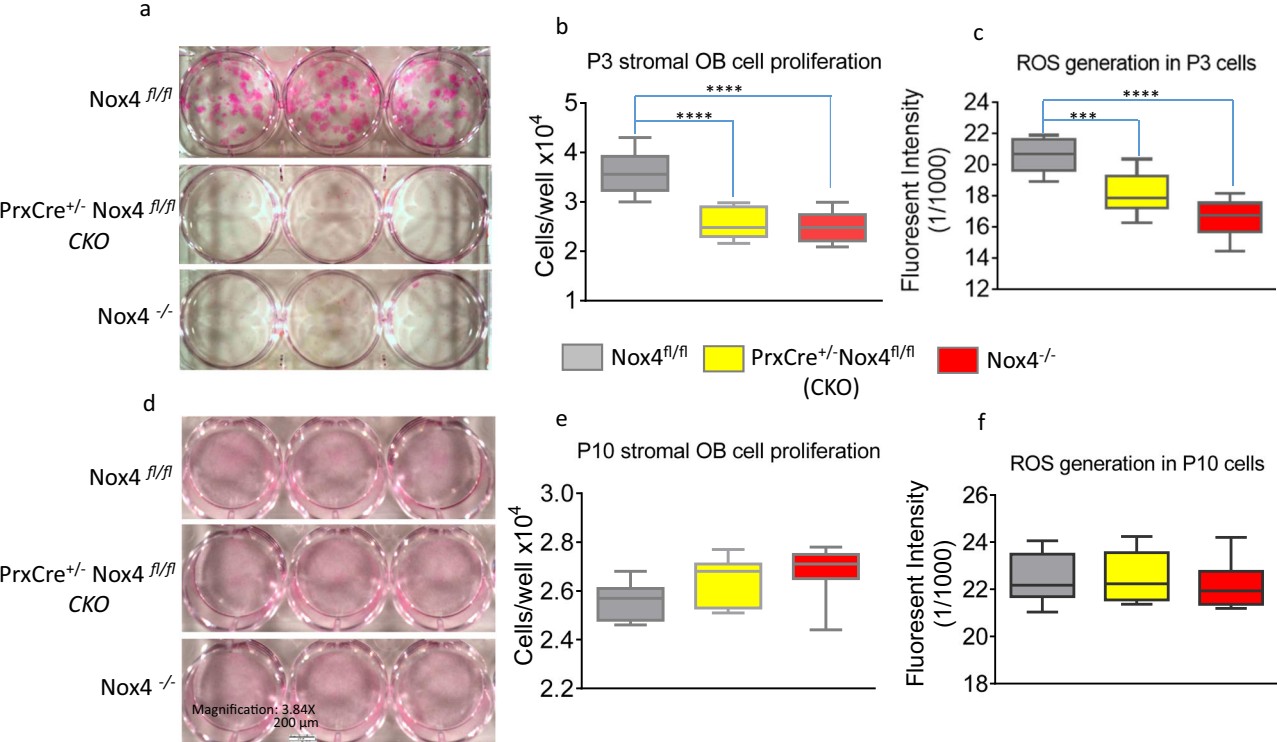

**Fig. 4 Cell proliferation and ROS generation of bone marrow stromal cells from Nox4-depleted mice catch up to cells from control mice after 10 passages. a** Representative pictures of hematoxylin blue stained cell culture wells of triplicates of passage 3 cells from *Nox4^{fl/fl}* control, *PrxCre^{+/-} Nox4^{fl/fl}* CKO, and *Nox4^{-/-}* mice. **b** Cell proliferation of cells from culture wells of triplicates of passage 3 cells from mice of the three genotypes. **c** ROS generation from cells of cultured wells of triplicates of passage 3 cells from mice of the three genotypes. **d** Representative pictures of hematoxylin blue stained cell culture wells of triplicates of passage 10 cells of the three Nox4 genotypes. **e** Cell proliferation of cells from culture wells of triplicates of passage 10 cells. **f** ROS generation from cells of cultured wells of triplicates of passage 10 cells. Significant differences indicated by ***$p < 0.001$; ****$p < 0.0001$, by one-way ANOVA followed by Tukey's post hoc test.

them when they got confluent every 7 to 8 days in 6-well plates. On passage 3, cells from CKO and *Nox4^{-/-}* mice still showed lower self-renewal capacity (Fig. 4a), and cell proliferations were significantly lower compared with those cells from *Nox4^{fl/fl}* control mice (Fig. 4b). ROS generation in the cell cultures derived from CKO and *Nox4^{-/-}* mice were also significantly lower as compared to cell cultures derived from *Nox4^{fl/fl}* control mice (Fig. 4c). However, at passage 10, self-renewal capacity of cells from CKO and *Nox4^{-/-}* mice showed no differences compared with those cells from *Nox4^{fl/fl}* control mice (Fig. 4d), and cell proliferations showed no significant differences between genotypes either (Fig. 4e). ROS generation of these passage 10 cells from CKO and *Nox4^{-/-}* caught up to that of control cells from *Nox4^{fl/fl}* mice (Fig. 4e).

**Nox4-deficient animals have less trabecular bone at 3 weeks of age, but not at 32 weeks of age.** Bone microarchitecture was evaluated by micro-CT in 3-week-old Nox4 CKO, *Nox4^{-/-}*, and *Nox4^{fl/fl}* control mice. Representative scans of distal femur of 3-week-old male and female mice are shown in Fig. 5a, b, k, l, and subsequent bone histology are shown in Fig. 5c, m for H&E staining and in Fig. 5d, n for TRAPase staining, respectively. TRAPase staining for osteoclasts showed no obvious differences between the genotypes. Among parameters that were analyzed on trabecular bone, we present percentage of trabecular bone volume BV/TV (bone volume/total tissue volume) (Fig. 5e, o), trabecular number (Tb.N) (Fig. 5f, p), BS/TV (bone surface/total tissue volume) (Fig. 5g, q), trabecular thickness (Tb.Th) (Fig. 5h, r), trabecular separation (Tb.Sp) (Fig. 5i, s) and bone mineral density

(BMD) (Fig. 5j, t) from male and female mice, respectively. There was a striking reduction in the amount of trabecular bone in the two Nox4-deficient genotypes. Relative bone volume and trabecular numbers were significantly lower in Nox4 CKO and *Nox4^{-/-}* mice than in *Nox4^{fl/fl}* mice in both sexes. There was slightly reduced BS/TV in the Nox4-deficient mice reaching statistical significance between *Nox4^{-/-}* and *Nox4^{fl/fl}* genotypes in males, and between CKO and *Nox4^{fl/fl}* genotypes in females. BMD also showed reduced levels in Nox4-deficient mice with statistical significance between CKO and *Nox4^{fl/fl}* genotypes in males, and between *Nox4^{-/-}* and *Nox4^{fl/fl}* genotypes in females. Finally, trabecular separation was significantly increased in CKO mice relative to control mice in both sexes. Since the reduction in trabecular bone occurs in the CKO genotype, we conclude that it is Nox4 in cells of the chondro/osteoblastic lineage that is responsible for this phenotype.

We have performed a similar micro-CT analysis of femurs isolated from 32-week-old mice. Representative scans of femurs are shown in Fig. 6a, b for males and Fig. 6k, i for females. Bone histology are shown in Fig. 6c for H&E staining and in Fig. 6d for TRAPase staining in males and in Fig. 6m for H&E staining and in Fig. 6n for TRAPase staining in females. Trabecular bone parameters are depicted in Fig. 6e–j in male and Fig. 6o–t in female. At this age, there was no diminished trabecular bone in Nox4-deficient animals compared to control mice. There was slightly higher trabecular thickness in female *Nox4^{-/-}* mice compared to *Nox4^{fl/fl}* mice. In males, BV/TV, BS/TV and Tb.N parameters were significantly higher in Nox4 CKO mice than in both *Nox4^{fl/fl}* and *Nox4^{-/-}* mice. In retrospect, this phenomenon of increased trabecular bone in only the male CKO mice can also be observed in micro-CT of tibias from

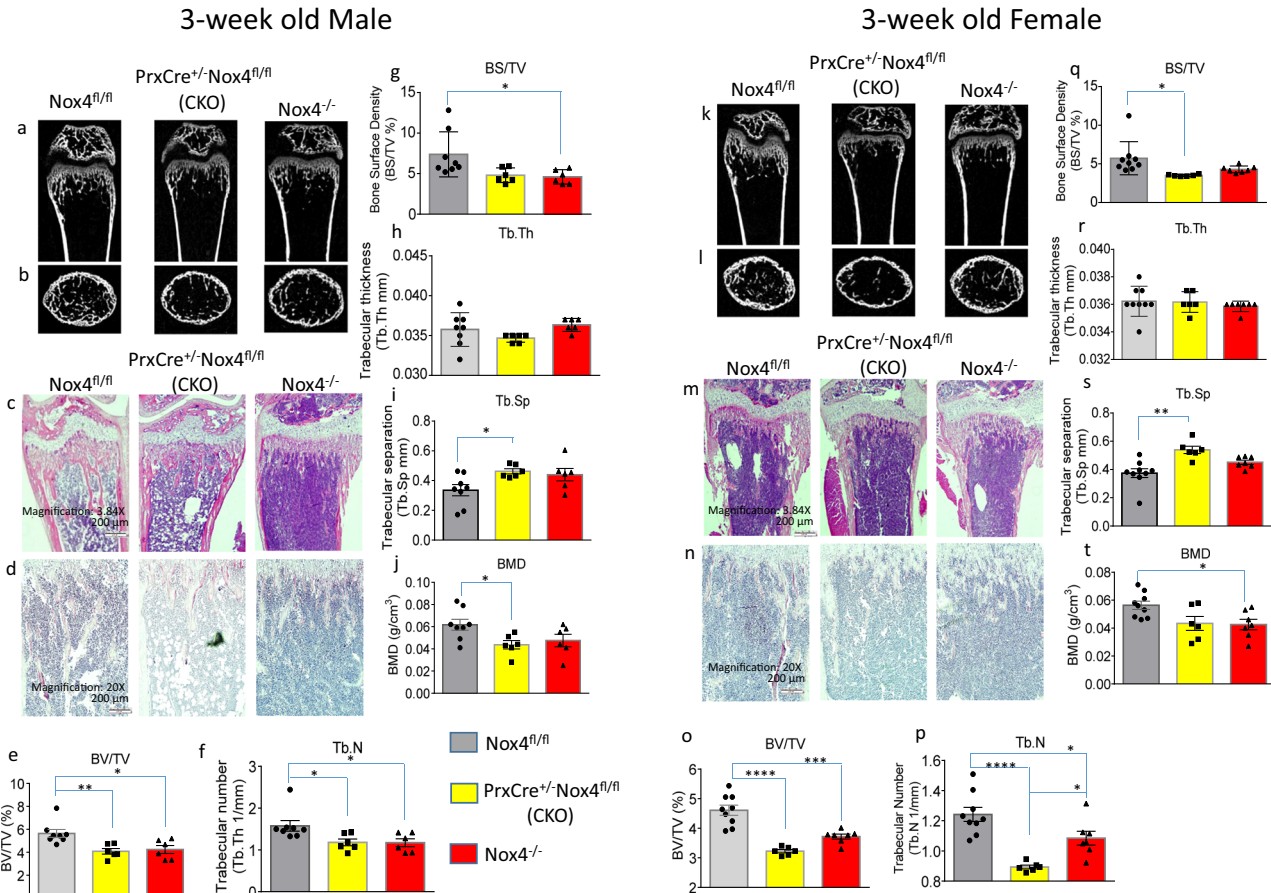

**Fig. 5 Micro-CT bone phenotype of femurs of 3-week-old *Nox4fl/fl* control, *PrxCre+/- Nox4fl/fl* CKO, and *Nox4-/-* mice. a**, **b** Representative micro-CT images of sagittal and transverse views of the distal femur from one sample from each group of male mice; upper panel shows sagittal view and lower panel shows transverse view. White lines and dots indicates trabecular or cortical bone tissues. **c**, **d** H&E and TRAPase histological stainings of the distal femur from one sample from each group of male mice. **e**–**j** Micro-CT measures of six parameters from trabecular femoral bone from male mice. BV/TV, bone volume/total tissue volume; BV, bone volume; BS, bone surface; BS/TV, bone surface density; Tb.N, trabecular number; Tb.Th, trabecular thickness; Tb.Sp, trabecular separation; BMD, connective tissue bone mineral density. **k**, **l** Representative micro-CT images of sagittal and transverse views of the distal femur from one sample from each group of female mice. **m**, **n** H&E and TRAPase histological stainings of the distal femur from one sample from each group of female mice. **o**–**t** Micro-CT measures of six parameters from trabecular femoral bone from female mice. Data are expressed as mean ± SD, analyzed by one-way ANOVA, additionally, *p < 0.05, **p < 0.01, ***p < 0.001, ****p < 0.0001 by Tukey's multiple comparison. There were 6–9 mice per group.

the same animals (35). To determine if this is a reproducible observation, we performed micro-CT of tibias from a new, independent set of 32-week-old animals (Supplemental Table 1). For this data set, BV/TV and BS/TV parameters are still higher in CKO than in *Nox4fl/fl* males, but there are no significant differences between the two Nox4-depletion genotypes. Trabecular numbers are not significantly different between genotypes, but trabecular thickness is significantly higher in both *Nox4-/-* and CKO males compared to *Nox4fl/fl* males. In total, Nox4 deficiency in cells of the chondro/osteoblastic lineage in 32-week-old males, but not females, is associated with increased trabecular bone in femurs and tibias. The Nox4 genotypes had no major effects on femoral cortical parameters at either age (Supplemental Tables 2 and 3). Immunostainings on 3-week-old and 32-week-old male and female femur histological sections for Prx1, Nox4 and Cathepsin k for osteoblastic and osteoclastic cells are shown in Supplemental Figs. 1 and 2.

**Normal trabecular bone levels are reached at 6 weeks of age in Nox4-deficient mice.** To determine how long depressed trabecular bone persists in Nox4-deficient mice, we performed micro-CT on femurs and analyzed samples from two additional ages:

6 weeks and 13 weeks. We did not observe significant differences in body weight and femoral length between the three Nox4 genotypes at any time point in either sex (Fig. 7a, b). Already at 6 weeks of age there were no longer significantly lower levels of trabecular bone as determined by parameters BV/TV and Tb.N (Fig. 7c, d). Enhanced levels of trabecular bone in male CKO or *Nox4-/-* mice as compared to *Nox4fl/fl* mice were not observed at 6 and 13 weeks of age (Fig. 7c). It can also be noticed that levels of trabecular bone at 32 weeks of age are decreasing relative to earlier time points. For both sexes, we did not observe significant differences of the presented femoral cortical parameters at any age (Fig. 7c, d). These data clearly indicate that Nox4 gene deletion, either osteoblastic cell-specific or systemically, affects trabecular bone development. We have presented complete sets of femoral micro-CT trabecular and cortical data of 6-week-old and 13-week-old mice of the three Nox4 genotypes in Supplemental Tables 4–7.

**The transcriptome of the bone marrow is strongly affected by age, but not the Nox4 genotype.** Since the bone marrow is a source of mesenchymal stem cells including osteoblastic

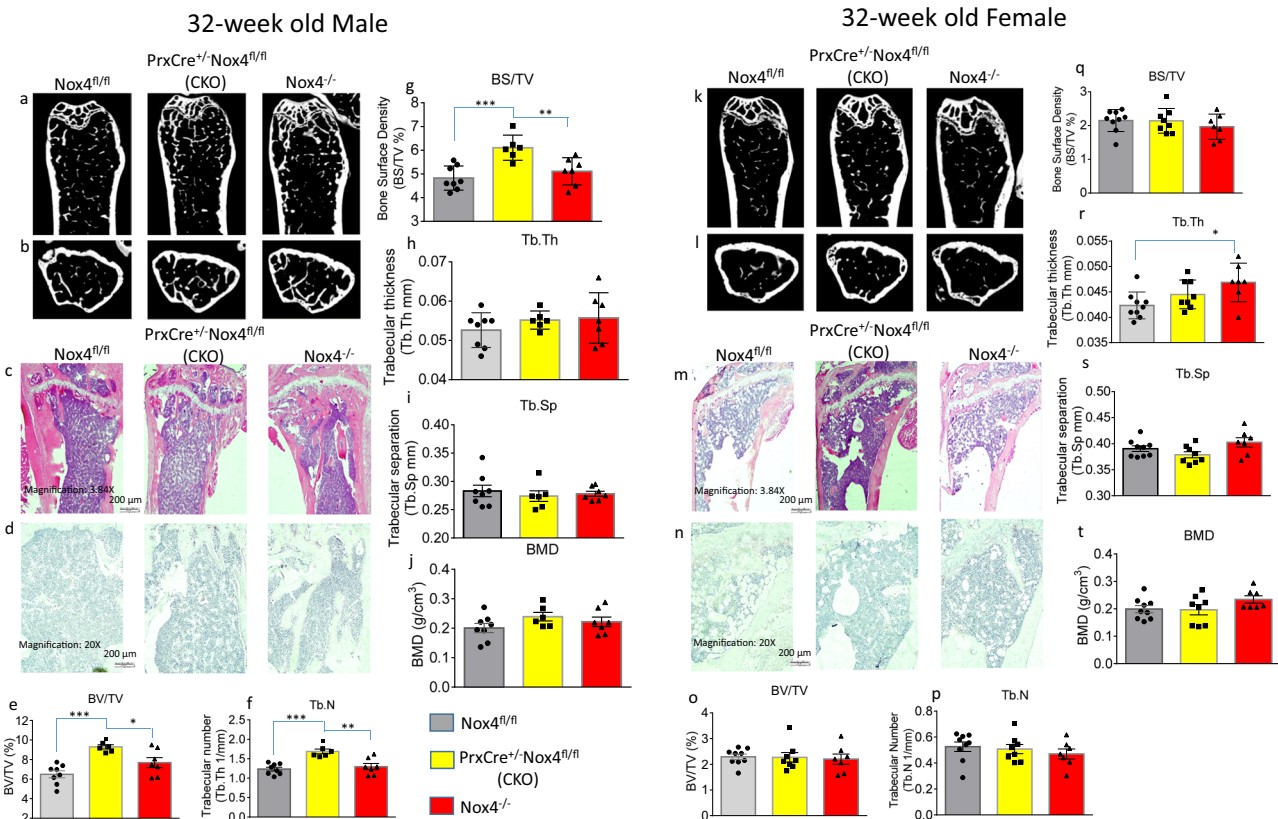

**Fig. 6 Micro-CT bone phenotype of femurs of 32-week-old *Nox4^fl/fl* control, and *PrxCre^+/-Nox4^fl/fl* CKO and *Nox4^-/-* mice. a**, **b** Representative micro-CT images of sagittal and transverse views of the distal femur from one sample from each group of male mice; upper panel shows sagittal view and lower panel shows transverse view. White lines and dots indicates trabecular or cortical bone tissues. **c**, **d** H&E and TRAPase histological stainings of the distal femur from one sample from each group of male mice. **e–j** Micro-CT measures of six parameters from trabecular femoral bone from 32-week-old male mice. BV/TV, bone volume/total tissue volume; BV, bone volume; BS, bone surface; BS/TV, bone surface density; Tb.N, trabecular number; Tb.Th, trabecular thickness; Tb.Sp, trabecular separation; BMD, connective tissue bone mineral density. **k**, **l** Representative micro-CT images of the distal femur from one sample from each group of female mice. **m**, **n** H&E and TRAPase histological stainings of the distal femur from one sample from each group of female mice. **o–t** Micro-CT measures of six parameters from trabecular femoral bone from 32-week-old female mice. Parameter BV/TV, BS/TV, Tb.N, Tb.Th, Tb.Sp, and BMD were presented. Data are expressed as mean ± SD, analyzed by one-way ANOVA, additionally, *$p < 0.05$, **$p < 0.01$, ***$p < 0.001$, by Tukey's multiple comparison. There were 6–9 mice per group.

precursors as well as precursors of osteoclasts, we performed RNA sequencing of femoral marrow RNA from 3- and 32-week-old mice. There was a very strong effect of the age on gene expression in the bone marrow (Supplementary Results). The sex had a much smaller effect and affected mostly the older mice. Nox4 expression was very low in the bone marrow, and the Nox4 genotype had small effects on the transcriptome. The only gene strongly affected by the genotype was *Cyp2s1* whose expression was upregulated in the CKO mice. However, that is a likely consequence of the PrxCre construct rather than knockout of Nox4 expression[23,24].

**Altered bone turnover caused by Nox4 depletion**. We finally evaluated systemic bone formation and resorption conditions in mice at 3 weeks and 32 weeks of age by measuring levels of serum bone remodeling markers. We analyzed bone-specific alkaline phosphatase (ALP) and procollagen type 1 N-terminal propeptide (P1NP) as bone formation markers, and carboxy-terminal crosslinked telopeptide of type 1 collagen (CTX1) and Cathepsin K as bone resorption markers. At 3 weeks of age, levels of both ALP and P1NP were significantly lower in Nox4 CKO and *Nox4^-/-* mice compared with Nox4^fl/fl control mice for both sexes (Fig. 8a, c). Lower mean levels of the bone formation markers for the *Nox4^-/-* than for Nox4 CKO genotype could indicate a total

lower bone formation when Nox4 is knocked out in the whole skeleton as compared to the partial depletion in the CKO genotype (Nox4 is only depleted in parts of the skeleton of CKO mice such as the long bones). For the resorption markers, CTX-1 levels were elevated in the Nox4-deficient mice in 3 weeks old males, but not females (Fig. 8e), while Cathepsin K concentrations did not differ significantly between the genotypes (Fig. 8g). At 32 weeks of age, the only significant difference in bone formation markers among genotypes is a higher level of P1NP in the Nox4-deficient males compared to control males (Fig. 8b, d). Bone resorption markers present a mixed scenario with decreased concentrations of CTX1 in *Nox4^-/-* mice than in controls and increased amounts of Cathepsin K in Nox4 CKO females compared to controls (Fig. 8f, h). The changes of bone resorption markers are difficult to interpret. They are not be likely to be a direct consequence of Nox4 depletion in osteoclasts but rather sequelae of changes of osteoblastic bone formation. We conclude that Nox4 depletion leads to reduced bone formation in 3-week-old mice, but elevated collagen type 1 synthesis in 32-week-old male mice.

## Discussion

In the present study, we investigated the role of Nox4 during early osteoblast differentiation and proliferation and in bone

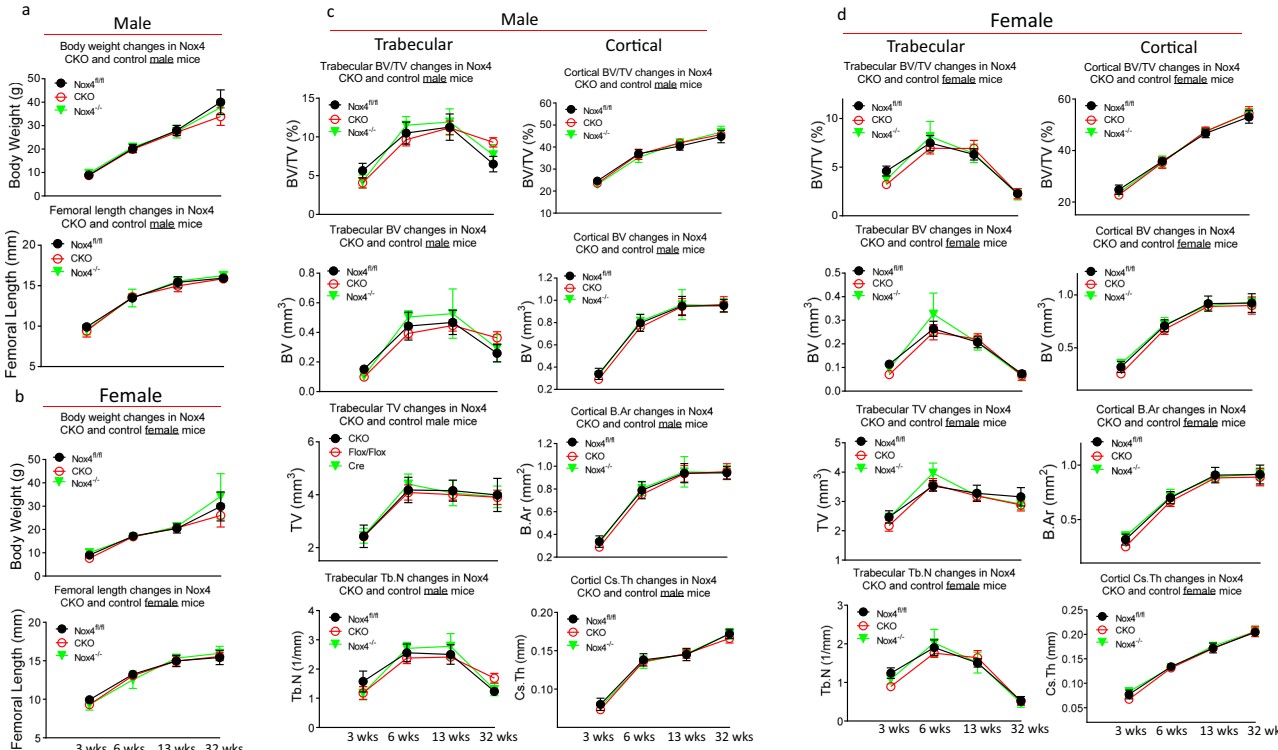

**Fig. 7 Time courses of body weight, femoral length, and femoral bone architecture.** Parameters were determined in $Nox4^{fl/fl}$, $PrxCre^{+/-} Nox4^{fl/fl}$ (CKO) and $Nox4^{-/-}$ mice at 3, 6, 13, and 32 weeks (wks) of age (**a**) Body weight and femoral length for male mice. **b** Body weight and femoral length for female mice. **c** Trabecular parameters BV/TV, BV, TV and Tb.N, and cortical parameters BV/TV, BV, B Ar (bone area), and Cs Th (cortical thickness) in male mice. **d** Trabecular parameters BV/TV, BV, TV and Tb.N, and cortical parameters BV/TV, BV, B Ar (bone area), and Cs Th (cortical thickness) in female. Means +/− SD are presented for 6–9 mice per group.

development from early life before weaning to the 32-week-old adult using global Nox4 knockout and pre-osteoblast-specific Nox4 gene deletion mouse models. Nox4 is robustly expressed in osteoblastic cells[18], but there is very little known about the role of Nox4 during osteoblast differentiation and maturation[25]. During osteoblastic differentiation of ST2 cells, Nox4 gene expression and ROS production were elevated during early bone cell differentiation while Nox1 and Nox2 expression rapidly declined to barely detectable levels. This indicates that Nox4 and its associated ROS signaling could be required for early osteoblast differentiation. This would be consistent with the requirements of Nox and ROS for development of other tissue types[26]. Indeed, stromal osteoblastic cells taken from either $Nox4^{-/-}$ or Nox4 pre-osteoblast CKO mice showed significantly lower potential for differentiation and proliferation/stem cell self-renewal. Furthermore, ROS production was significantly lower during early passages in stromal cell replicative cultures. We have also previously reported that knockdown of Nox4 in calvarial cells by shRNA reduce differentiation[19]. Although SAβG activity was lower in osteoblastic cells without Nox4 gene expression, cell proliferation was also significantly lower indicating Nox4-associated lower level of SAβG activity does not play a role in regulation of proliferation during early osteoblast differentiation. However, after stromal cells were passaged up to eight times, differentiation and proliferation/stem cell self-renewal of those stromal cells taken from either $Nox4^{-/-}$ or Nox4 pre-osteoblast CKO mice clearly caught up to control cells. These results suggest that Nox4 gene expression plays a role in promoting early osteoblast differentiation. Nox4 gene expression apparently is required for early life bone development (prior to weaning), but it seems to play less of a role in bone remodeling during adulthood as previously noted[20,23].

We have previously shown that Nox2 activity and signaling in the bone, which occurs mainly in macrophages and osteoclasts was markedly reduced in $p47^{phox-/-}$ mice, as indicated by decreased ROS generation in bone marrow cells. Osteoblastic cells from $p47^{phox-/-}$ mice had little Nox2 expression and failed to respond the Nox2 activator PMA[19]. Our previous research indicated that $p47^{phox-/-}$ mice represented systemic Nox2 gene knockout mouse model. We observed increased bone formation during development in early life of $p47^{phox-/-}$ mice that was reversed in old $p47^{phox-/-}$ mice (2 years old) compared to their respective wild type controls. We concluded that $p47^{phox}$/Nox2 knockdown-mediated reduction of ROS generation appears only temporally beneficial for early bone development. Depletion of Nox4 had the opposite effect on bone formation in young mice: At 3 weeks of age, there is markedly reduced bone formation rates and reduced femoral trabecular bone. However, Nox4 gene expression is not required for young adult bone development, as trabecular bone is normal at 6 weeks of age. At 32 weeks of age, males might furthermore have enhanced trabecular bone and collagen synthesis when Nox4 is depleted. The age-related switch of bone mass in $p47^{phox-/-}$ mice appears to occur through an increased inflammatory milieu in bone accompanied by accelerated senescent signaling pathway in osteoblasts and that $p47^{phox}$/Nox2-dependent physiological ROS signaling suppresses inflammation in aging bone[19]. However, age-dependent changes of bone phenotype in $Nox4^{-/-}$ and Nox4 pre-osteoblast CKO compared with their wild type control mice were apparently due to significant changes of bone remodeling status. Our previous data in aged $p47^{phox-/-}$ mice were consistent with evidence that showed age-dependent increases in development of inflammatory arthritis in an entirely Nox2-deficient mouse model[11], which might conflict with studies suggesting a critical involvement of

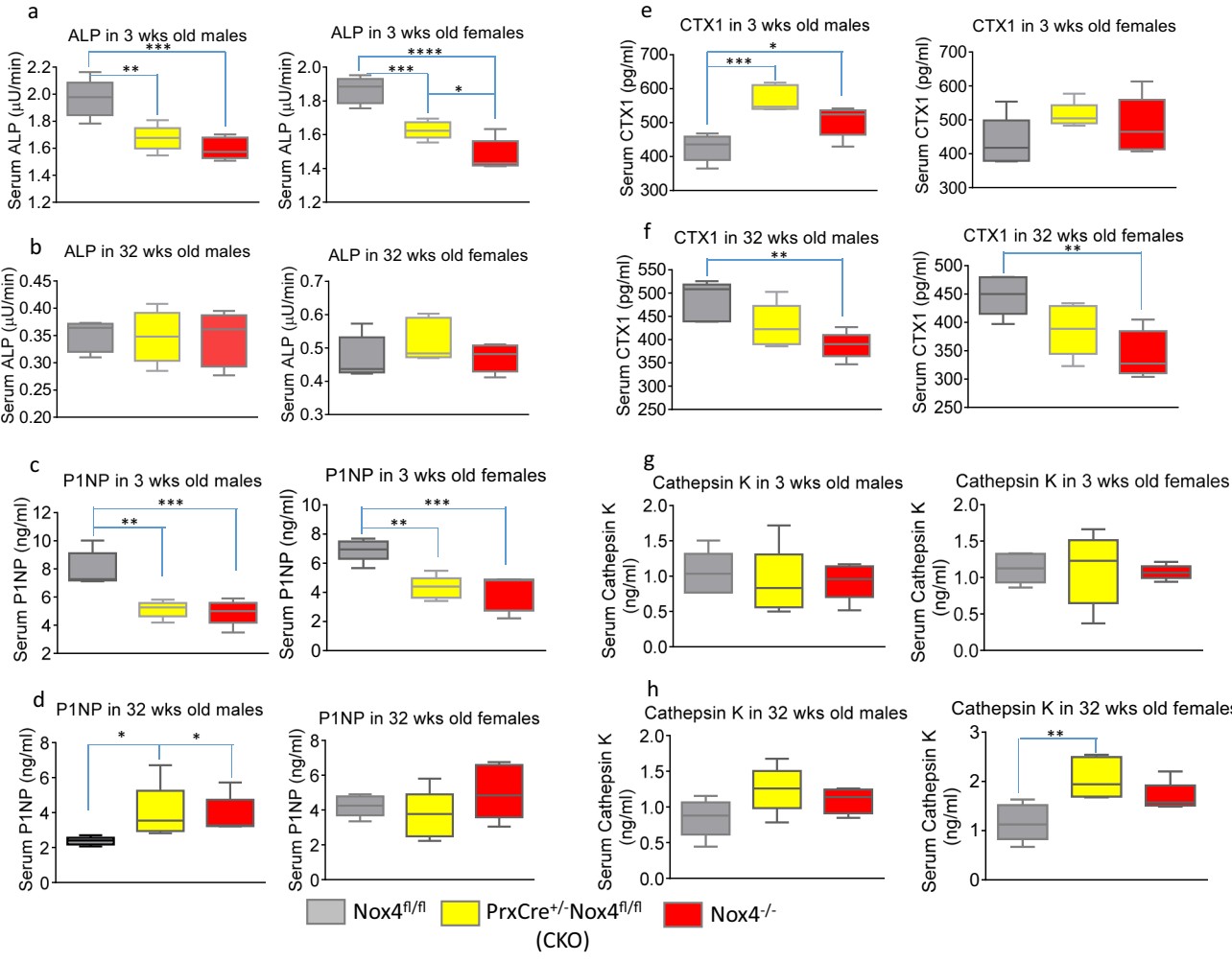

**Fig. 8 Bone remodeling markers in 3-week and 32-week-old *Nox4^fl/fl* control, *PrxCre^+/−* *Nox4^fl/fl* CKO, and *Nox4^−/−* mice.** Bone remodeling markers in serum were determined. **a** Serum ALP (bone-specific alkaline phosphatase) levels in 3 weeks (wks) old mice. **b** Serum ALP levels in 32 wks old mice. **c** Serum P1NP levels in 3 wks old mice. **d** Serum P1NP levels in 32 wks old mice. **e** Serum CTX1 levels in 3 wks old mice. **f** Serum CTX1 levels in 32 wks old mice. **g** Serum Cathepsin K levels in 3 wks old mice. **h** Serum Cathepsin K levels in 32 wks old mice. Data are expressed as mean ± SD, analyzed by one-way ANOVA, additionally, *$p < 0.05$, **$p < 0.01$, ***$p < 0.001$, ****$p < 0.0001$ by Tukey's multiple comparison.

ROS in age-associated bone loss and inflammation. Our current results in bone cells and bone development from *Nox4^−/−* and Nox4 pre-osteoblast CKO mice were consistent with the roles of Nox and ROS on cellular development in other tissues[27,28].

ROS production or accumulation are commonly thought to be toxic byproducts of cellular function, but specific Nox-dependent ROS production is not recognized as a byproduct[29]. Evidence suggested that Nox-dependent ROS signaling plays an important role on cell cycle progression and proliferation[13,30]. This was consistent with our current data suggesting that Nox4 expression and ROS production declines during late stages of osteoblast differentiation, and these declines were accompanied by significant decreases of Col1a1 and osteocalcin expression. Nox4 expression was abundant and Nox2 was also expressed in mature osteoblasts but there was very low expression of Nox1 and no expression of Nox3 were observed in these cells[18]. Global Nox4 knockout animals were previously noted to have increased femoral trabecular bone without changes in bone formation rates[26]. As our results demonstrate, the bone phenotype is crucially dependent on the age and to a lesser extent the sex of the mice. A role for Nox4 in early bone development is consistent with findings in other tissues demonstrating that Nox4 was required for cardiomyocyte and adipocyte differentiation in vitro

via redox activation[31]. Our data also supported the notion that Nox4-associated ROS-dependent signaling pathways are importantly playing a role on osteoblast differentiation[25].

The bone marrow is a source of both mesenchymal stem cells and hemopoietic precursors of osteoclasts. We observed that the whole bone marrow transcriptome at 3 weeks of age was markedly different from the transcriptome at 32 weeks of age. Despite Nox4 expression being critical for trabecular bone formation at 3 weeks of age, we could not detect clear effects of the Nox4 genotype on genes important for bone turnover in the marrow compartment. That may be a consequence of Nox4-expressing mesenchymal stem cells being vastly outnumbered by cells of the hemopoietic lineage without Nox4 expression. Further research could be directed at the role of Nox4 specifically in the mesenchymal stem cells early in life.

Early life bone development and skeletal remodeling during adults are two different processes. The current understanding on bone development throughout life is that increases in bone formation relative to bone resorption in rapidly growing rodents results in bone accrual. The function of osteoclast bone resorption is mainly to keep bone in fine shape during early development, although decreases in osteoclast bone resorption could result in increased bone mass with uncertain quality[32]. On the other hand,

decreases in bone formation and osteoblast function in addition to increases in bone resorption with un-balanced bone remodeling are considered features of bone loss in adults[5]. Although osteocytes were suspected to be involved in bone remodeling[33,34], active osteoblasts are still recognized as key resources to maintain appropriate bone formation rate in adults[4]. While Nox-associated ROS production, defective oxidative stress and tissue cell proliferation are very complex mechanisms[28], increasing Nox-associated ROS production might not be harmful for somatic cell proliferation during adult life[27,35,36].

The major conclusions from this work are that Nox4 expressed in the chondro-osteoblastic lineage promotes bone formation in young mice prior to weaning and that Nox4 promotes osteoblastic cell self-renewal. To dissect if Nox4 signaling itself or its associated ROS signaling determine bone cell differentiation and proliferation, it is important to consider different approaches to manipulation of Nox gene expression and types of ROS produced and to target these processes in young versus adult mice.

## Methods

**Materials and reagents**. Commercially available materials and reagents are listed in Supplementary Table 8, and primer sequences used for real-time PCR are listed in Supplementary Table 9.

**Animals**. Nox4-floxed mice of genotype *Nox4fl/fl* and mice of genotype *PrxCre+/- Nox4fl/fl* with conditional knockout (CKO) of Nox4 in the limb bud mesenchyme of long bones were described previously[23,26]. Nox4 whole-body knockout mice of genotype *Nox4 0/0* lacking exons 1 and 2 were previously generated (labeled *Nox4 0/0* in reference[23]). Mice were maintained on standard rodent chow until euthanasia by $CO_2$ asphyxiation. Mice at 13 and 32 weeks of age, for which femoral data are presented in Figs. 6 and 7 were the same mice used for micro-computed tomography of tibia in reference[23]. The Institutional Animal Care and Use Committee of Louisiana State University Health Sciences Center approved all animal experiments in accordance with the Guide for the Care and Use of Laboratory Animals (U.S. National Institutes of Health).

**Cell cultures**. Bone marrow stromal and non-adherent bone marrow hematopoietic cells were isolated from 18 to 31 weeks old female *PrxCre+/- Nox4fl/fl*, *Nox4-/-*, and *Nox fl/fl* mice; three mice per group for in ex vivo cultures. For osteoblast stem cell self-renewal, we cultured $2 \times 10^5$ stromal cells/well in a six-well plate. When the cells became ~80% confluent after 6 days, they were passaged. This procedure was repeated 11 times for a total of 10 passages. Bone marrow-derived mouse stromal cell line ST2 cells were purchased from the Riken Cell Bank (Ibaraki, Japan)[37]. Murine macrophage cell line RAW264.7 cells (from American Type Culture Collection) and non-adherent bone marrow cells were cultured in 96-well plates ($2 \times 10^4$ cells/well) in the presence or absence of 30 ng/ml of RANKL, in α-MEM supplemented with 10% fetal bovine serum, penicillin (100 units/ml), streptomycin (100 μg/ml), and glutamine (4 mM). TRAPase pink staining for assessing osteoclastic cell differentiation and activity was performed based on a method published previously[38].

**Measurement of ROS**. Cell-based ROS/superoxide detection assay kit was purchased from Abcam (abcam.com), cat#ab139476. According to manufactory's recommendation, this assay Kit was designed to directly monitor real-time global levels of total ROS production in live cells using a microplate reader at Ex/Em 490/525 nm. Detailed measurement procedure can be found from in the protocol provided by manufacturer.

**Cell proliferation assay**. Non-Radioactive cell proliferation assay was performed following the protocol provided by manufacturer (Promega Corporation. Part#TB169). According to the manufacturer's instruction, this assay measures absorbance at 490 nm, and there is a linear correlation (the correlation coefficient of the line is 0.997) between cell number and absorbance at 490 nm, absorbance was converted to cell number.

**Bone analyses**. At sacrifice, femurs were removed and stored at −80 °C. Tibiae were fixed in 10% neutral buffered formalin. Micro-computed tomography (micro-CT) measurements of the trabecular and cortical compartments from the left femur were evaluated using SkyScan μCT scanner (SkyScan 1272, Bruker.com) at 8 μm pixel size with X-ray source power of 60 kV and 166 μA and integration time of 950 ms. The trabecular compartment included a 0.9 mm region extending distally 0.03 mm from the physis. The grayscale images were processed by using a filter (=Al, $\sigma = 0.5$, mm) to remove noise, and a fixed threshold of 125 was used to extract the mineralized bone from the soft tissue and marrow phase. Cancellous

bone was separated from the cortical regions by semi-automatically drawn contours. A total of 120 slices starting from about 0.1 mm distal to growth plate, constituting 0.80 mm length, were evaluated for trabecular bone structure. Micro-CT parameters, including bone volume fraction (BV/TV, %), trabecular thickness (Tb.Th, mm), trabecular separation (Tb.Sp, mm), trabecular number (Tb.N, 1/mm) and bone mineral density (BMD) were analyzed based on description by Bouxsein et al.[39], and by using software provided by SkyScan, Bruker. For cortical bone, the cortical compartment was a 0.6 mm region extending distally starting 5 mm proximal to the knee joint. Total cross-sectional area (CSA, mm²), medullary area (MA, mm²) and cortical thickness (Ct.Th, mm) were assessed[40–43].

**Serum bone turnover markers measurement**. Serum were collected at the time of animal sac and tissue harvest. The serum P1NP (Procollagen 1 N-terminal Propeptide) levels were measured by direct immunoassay P1NP assay Kit. The P1NP level measurement ELISA kit was purchased from Mybiosource.com (Catalog No: MBS2500076) and measurement procedure followed the manufacturer's recommendations. The serum bone formation marker alkaline phosphatase (ALP) and the serum bone resorption marker C-terminal telopeptides of type I collagen (CTX1) RatLaps were measured by Rat-MID™ ALP ELISA and RatLaps™ ELISA, respectively, from Nordic Biosciences Diagnostic (Herlev, Denmark). Serum Cathepsin K levels were measured by an ELISA-based kit from MyBioSource (mybiosource.com), cat#MBS164601.

**Real-time reverse transcription–polymerase chain reaction (RT-PCR) analysis**. RNA was extracted using TRI Reagent (MRC Inc., Cincinnati, OH) according to the manufacturer's recommendation followed by DNase digestion and column cleanup using QIAGEN mini columns. RNA isolation from cultured cells: treated cells from 12-well plates were washed twice with PBS, 1000 μl TRI Reagent was added into each well. Cells were scraped into a 1.5-ml Eppendorf tube. RNA preparation was the standard TRI Reagent protocol. Reverse transcription was carried out using an iScript cDNA synthesis kit from Bio-Rad (Hercules, CA). Real-time RT-PCR was carried out using SYBR Green and an ABI 7500 Fast sequence detection system (Applied Biosystems, Foster City, CA).

**Senescence-associated β-galactosidase (SAβG) activity assay**. SAβG activity assay was performed by β-galactosidase enzyme assay kit (Promega) measured the absorbance at 420 nm according to manufacturer's instruction as we described previously[(19,20)].

**RNA sequencing (RNA-Seq)**. Bone marrow was released from femurs by centrifugation at $5700 \times g$ for 30 s into RNAlater followed by TRI Reagent lysis and RNA isolation by the TRI Reagent protocol[44]. RNA-Seq was conducted for marrow RNA isolated from male and female mice of the three Nox4 genotypes at 3 and 32 weeks of age. There were three samples per combination of sex, genotype, and age. Each sample was marrow RNA pooled from 2 to 4 mice. RNA-Seq was conducted on the Illumina NextSeq 500 platform. Alignment and analysis was done in Partek Flow as follows: Contaminants were removed with Bowtie 2 v2.2.5 and reads were aligned to mm10 using STAR v2.7.3a and quantified using RefSeq Transcripts v99 (released 2021-08-02). Normalization was done with Trimmed Mean of M-values (TMM). Normalized counts were calculated as log2 (TMM + 1). Analysis of differences between groups were based on normalized counts using analysis of variance (ANOVA) and DeSeq2. RNA-Seq data are deposited in the Gene Expression Omnibus database as GEO submission number GSE195454.

**Statistics and reproducibility**. Statistical power was computed based on a two-factor ANOVA with 7–9 mice per group. Statistical analysis was performed with GraphPad Prism 8.0 (GraphPad Software, Inc., San Diego, Ca, USA). Numerical variables were expressed as means ± STDEV (Standard Deviation). Comparisons between groups were performed with the nonparametric Kruskal–Wallis test followed by a Dunnett's test comparing each dose to the control group. The nonparametric Wilcoxon rank-sum test was used to compare controls to individual treatment. The rationale for using 7–9 mice per group is described in the Supplementary Methods. Cell culture experiments were conducted at least three independent times, and representative images are displayed. T-test, one-way analysis of variance (ANOVA) followed by Tukey's multiple comparisons test post hoc analysis was used to compare different genotypic groups of mice. The critical p-value for statistical significance was $p = 0.05$.

**Reporting summary**. Further information on research design is available in the Nature Research Reporting Summary linked to this article.

## Data availability

Source data underlying plots shown in figures are available in excel sheet of Source Data 1 and are shown in Supplementary information. Additional data related to the paper are available from the corresponding author on reasonable request. RNA-Seq data were deposited in the Gene Expression Omnibus database as GEO submission number GSE195454.

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

## Acknowledgements

This work was supported in part by National Institute of Health grant R37 AA18282 (M.J.J.R.) and supported by USDA-ARS Project 6026-51000-012-06S to Arkansas Children's Nutrition Center sub-award to J.R.C.

## Author contributions

Study conceived and planned by J.R.C. and J.R.C. wrote the paper. J.R.C. and M.J.J.R. are senior authors designed and performed the study; cell, biochemical, and molecular in ex vivo and in vitro experimental work by J.R.C., O.P.L., M.L.B., C.E.R., J.Z., K.S., K.B.P., data analysis by J.R.C., J.F.C., K.B.P, J.Z.; manuscript composition by J.R.C., J.F.C.; all authors discussed the results and edited the manuscript.

## Competing interests

The authors declare no competing interests.
