## [Peer Review File · Communications Biology]

Reviewers' comments:

Reviewer #1 (Remarks to the Author):

In this study, the authors claimed that NOX4 expression is essential during early age bone remodeling whereas not essential for late age bone remodeling. This is a very interesting and novel finding but surprising! Overall, this a well-written manuscript. However, the following concerns should be addressed.

Major:

1. The impact of Nox4 deletion on bone development during early age and late is determined. It is surprising why the increase in ROS production in the early stage followed by a decrease in ROS production in the late stage increase both osteoblastogenesis and osteoclastogenesis.
2. Conclusion: "These data suggest that Nox4 expression or ROS signaling in bone and osteoblastic cells play different roles during osteoblast differentiation, proliferation and maturation" not quite clear
3. Osteoclast and osteoblast staining of bone sections would be informative.
4. Molecular mechanisms are not clear. To determine the molecular mechanisms of Nox4 mediated modulation of osteogenesis during early and later periods of life, a gene array study would be useful.

Minor:

Few typographical errors are there.

"Our data also supported the notion that Nox4-associated ROS-dependent signaling pathways are important role in osteoblast differentiation" correct this sentence.

Reviewer #2 (Remarks to the Author):

Authors have used Nox4 global and conditional knockout mice, as well as cell lines, to show roles for Nox4/ROS during osteoblast proliferation and differentiation. Specifically, Nox4^{-/-} and CKO mice appear to have reduced bone formation and mass in younger ages.

Overall, the methodology is solid and suitable. There are however areas in which the manuscript could be improved and further validation of the cell culture experiments could be achieved by histology of the samples used for microCT analyses (for example, TRAP staining for osteoclasts).

Specific points/questions for authors:

- Introduction, paragraph 2 line 12 – Authors state 'several reports' suggesting... but then only reference one. If there are several reports, I would suggest referencing more of them.
- Figure 1 (and subsequent figures) – need to be clearer about significance. Legend states that * indicates $p < 0.05$, but is that significantly greater than Day 1, or less than Day 1? If significance has been based on vs Day 1, is there difference between other days, as this may also be relevant.
- Results, page 5, second to last line – authors state that Nox4^{-/-} and CKO stromal cells are growing slower. Can this conclusion be made from staining for CFU? It may mean that the cells are less proliferative, but not sure the conclusion can be made from this experiment that they are slower.
- Figure 2, panel A – the images are a little small and hard to see. Maybe just select a representative from each group and make it larger, rather than having the three. Also, the mouse numbers (95-3 etc.) are a little confusing. I think for publication purposes it may be better to simplify to mouse 1, 2, 3 or similar.
- Results, page 6, end of first paragraph – authors note that the finding of decreased SAbG expression was interesting, why is it interesting? Was this unexpected? Further discussion of why this is interesting (in discussion section) would be useful.
- Results/Figure 3 – what parameter was used to distinguish an osteoclast (i.e. 3 or more nuclei)? It looks from the representative images that the osteoclasts in the fl/f; mice are larger. Could it be that while total number of osteoclasts didn't differ, the size or number of nuclei did?
- Figure 5 – graphs need a figure legend and description of shapes/colours in legend.
- Figure 5 – representative images of CT scans are difficult to see due to size, can they be made

larger?

- Materials and methods, cell cultures – please include a brief description of how the bone marrow cells are isolated from mice
- Materials and methods, cell cultures – please include a brief description of how the standard ALP staining was performed
- Materials and methods, cell cultures – what does ACCT stand for? Please spell out in full in the first instance
- Discussion – considerable attention is given to the differences in phenotype between Nox2 mice from a previous study and Nox4 in the current study, however to a non-Nox researcher the relevance of this is unclear. What are the differences/similarities between Nox2 and 4? Would you expect to have similar or different phenotypes?

Reviewer 1

Reviewer: **In this study, the authors claimed that NOX4 expression is essential during early age bone remodeling whereas not essential for late age bone remodeling. This is a very interesting and novel finding but surprising! Overall, this a well-written manuscript. However, the following concerns should be addressed.**

Author's response: We thank this reviewer very much for his/her encouraging comments, we have provided additional data to address reviewer's concerns in our revised manuscript.

Reviewer: **The impact of Nox4 deletion on bone development during early age and late is determined. It is surprising why the increase in ROS production in the early stage followed by a decrease in ROS production in the late stage increase both osteoblastogenesis and osteoclastogenesis.**

Author's response: Although our statement in the text might be confusing, that was the key piece of evidence we presented in our manuscript. We have revised the statement as follows : physiological levels of Nox4-dependent ROS production in early life stages appear to be needed for stimulation of both osteoblastogenesis and osteoclastogenesis. However, elimination of accumulated ROS production in aging also appears to be required for keeping appropriate balance of osteoblastogenesis and osteoclastogenesis in older animals.

Reviewer: **Conclusion: "These data suggest that Nox4 expression or ROS signaling in bone and osteoblastic cells play different roles during osteoblast differentiation, proliferation and maturation" not quite clear.**

Author's response: We have clarified this sentence as follows: These data suggest that Nox4 expression and ROS signaling in bone and osteoblastic cells coordinately play an important role in osteoblast differentiation, proliferation and maturation (page 2, line 20-22).

Reviewer: **Osteoclast and osteoblast staining of bone sections would be informative.**

Author's response: We have performed bone histology, new data supporting our hypothesis are presented in Figure 5 and 6 and supplemental Figure 1 and 2 of revised manuscript.

Reviewer: **Molecular mechanisms are not clear. To determine the molecular mechanisms of Nox4 mediated modulation of osteogenesis during early and later periods of life, a gene array study would be useful.**

Author's response: We have performed RNA Seq of bone marrow cells isolated from all groups of 3 week and 32 week old male and female mice. A large data set suggesting potential molecular mechanisms of Nox4 mediated modulation of

osteogenesis during early and later periods of life is now presented in the Supplemental word document of the revised manuscript.

Reviewer: **Minor:**

Few typographical errors are there.

“Our data also supported the notion that Nox4-associated ROS-dependent signaling pathways are important role in osteoblast differentiation” correct this sentence..

Author’s response: We have carefully revised the manuscript and correct those typographical errors. We correct the sentence as: Our data also supported the notion that Nox4-associated ROS-dependent signaling pathways play an important role in osteoblast differentiation (Page 11, line 24-26).

Reviewer: 2

Reviewer: Authors have used Nox4 global and conditional knockout mice, as well as cell lines, to show roles for Nox4/ROS during osteoblast proliferation and differentiation. Specifically, Nox4^{-/-} and CKO mice appear to have reduced bone formation and mass in younger ages. Overall, the methodology is solid and suitable. There are however areas in which the manuscript could be improved and further validation of the cell culture experiments could be achieved by histology of the samples used for microCT analyses (for example, TRAP staining for osteoclasts).

Author's response: We thank this reviewer for spending his/her important time reviewing our work, and giving encouraging comments. We have taken reviewer's suggestions and performed bone histology in the revision, Figure 5 and 6 and supplemental Figure 1 and 2 of revised manuscript.

Reviewer: Introduction, paragraph 2 line 12 – Authors state ‘several reports’ suggesting... but then only reference one. If there are several reports, I would suggest referencing more of them.

Author's response: We have revised it according to reviewer's suggestion.

Reviewer: Figure 1 (and subsequent figures) – need to be clearer about significance. Legend states that * indicates $p < 0.05$, but is that significantly greater than Day 1, or less than Day 1? If significance has been based on vs Day 1, is there difference between other days, as this may also be relevant.

Author's response: * indicates $p < 0.05$, that means there is a significant difference from Day 1. However, it appears that Nox4, Nox1 and Nox2 mRNA expressions are time-dependent.

Reviewer: Results, page 5, second to last line – authors state that Nox4^{-/-} and CKO stromal cells are growing slower. Can this conclusion be made from staining for CFU? It may mean that the cells are less proliferative, but not sure the conclusion can be made from this experiment that they are slower.

Author's response: The proliferation of Nox4^{-/-} and CKO stromal cells are obviously and significantly decreased (Figure 2D), we have made changes of this statement in the revision.

Reviewer: Figure 2, panel A – the images are a little small and hard to see. Maybe just select a representative from each group and make it larger, rather than having the three. Also, the mouse numbers (95-3 etc.) are a little confusing. I think for publication purposes it may be better to simplify to mouse 1, 2, 3 or similar.

Author's response: Thanks for reviewer's suggestions, the images just show blue-pink colonies, we have numbered 1,2 ,3 as the mouse numbers in the revision.

Reviewer: Results, page 6, end of first paragraph – authors note that the finding of decreased SAbG expression was interesting, why is it interesting? Was this unexpected? Further discussion of why this is interesting (in discussion section) would be useful.

Author's response: Actually, it is interesting, we have added to the discussion to address this finding better.

Reviewer: Results/Figure 3 – what parameter was used to distinguish an osteoclast (i.e. 3 or more nuclei)? It looks from the representative images that the osteoclasts in the fl/f; mice are larger. Could it be that while total number of osteoclasts didn't differ, the size or number of nuclei did?

Author's response: Osteoclasts were identified as TRAPase positive with 3 or more nuclei per cell, it is possible that the size of osteoclasts might be different between groups we have added this possibility to the results.

Reviewer: Figure 5 – graphs need a figure legend and description of shapes/colours in legend.

Author's response: We have done in revision.

Reviewer: Figure 5 – representative images of CT scans are difficult to see due to size, can they be made larger?

Author's response: Micro-CT scan image shows white lines and dots for trabecular or cortical bone, we presented the standard size for publication purposes.

Reviewer: Materials and methods, cell cultures – please include a brief description of how the bone marrow cells are isolated from mice.

Author's response: We have described brief process for isolation of bone marrow cells in the revision.

Reviewer: Materials and methods, cell cultures – please include a brief description of how the standard ALP staining was performed.

Author's response: Yes, we have provided brief description.

Reviewer: Materials and methods, cell cultures – what does ATCC stand for? Please spell out in full in the first instance.

Author's response: We have provide full name of ATCC (American Type Culture Collection).

Reviewer: Discussion – considerable attention is given to the differences in phenotype between Nox2 mice from a previous study and Nox4 in the current study, however to a non-Nox researcher the relevance of this is unclear. What are the differences/similarities between Nox2 and 4? Would you expect to have similar or different phenotypes?

Author's response: Relative expressions of Nox2 and 4 in different cell types are different and they appear to signal via different MAP kinase cascades. We expect their expression palys slightly different roles in different tissues. It will be difficult to dissect this unless NOXs are knocked out in the same sex and age matched.Cre mouse model, to compare bone phenotypes.

We thank all the reviewers for their time and thoughtful comments. We believe that the resultant revisions have significantly strengthened the manuscript.

Yours Sincerely

Jin-Ran Chen, M.D., Ph.D.
Associate Professor
College of Medicine
Department of Pediatrics
University of Arkansas for Medical Sciences
Arkansas Children's Nutrition Center
Arkansas Children's Research Institute

REVIEWERS' COMMENTS:

Reviewer #1 (Remarks to the Author):

Overall the revised manuscript covers the essential experiments to support their hypothesis. I recommend the revised manuscript to be accepted.

However, a note for future work: whole BM is not a right sample to perform gene array to determine gene modulation during osteogenesis. They should use Calvaria Bone Transcriptome instead. They can use BMM for osteoclastogenesis gene modulation as well.